# Identification of a BAHD Acyltransferase Gene Involved in Plant Growth and Secondary Metabolism in Tea Plants

**DOI:** 10.3390/plants11192483

**Published:** 2022-09-22

**Authors:** Shirin Aktar, Peixian Bai, Liubin Wang, Hanshuo Xun, Rui Zhang, Liyun Wu, Mengdi He, Hao Cheng, Liyuan Wang, Kang Wei

**Affiliations:** 1Key Laboratory of Tea Biology and Resources Utilization, Ministry of Agriculture, National Center for Tea Improvement, Tea Research Institute Chinese Academy of Agricultural Sciences (TRICAAS), Hangzhou 310008, China; 2Graduate School of Chinese Academy of Agricultural Sciences, Beijing 100081, China

**Keywords:** BAHD acyltransferase *(TEA031065)* gene, *Camellia sinensis*, *Arabidopsis thaliana*, secondary metabolites, gene expression

## Abstract

Plant acyl-CoA dominated acyltransferases (named BAHD) comprise a large appointed protein superfamily and play varied roles in plant secondary metabolism like synthesis of modified anthocyanins, flavonoids, volatile esters, etc. Tea (*Camellia sinensis*) is an important non-alcoholic medicinal and fragrancy plant synthesizing different secondary metabolites, including flavonoids. In the tea (*C.A sinensis*) cultivar Longjing 43 (LJ43), eight samples were performed into three groups for transcriptome analysis under three biological replications. Among the BAHD acyltransferase genes in tea cultivars, the expression of *TEA031065* was highest in buds and young leaves following the RNA sequencing data, which was coincident with the tissue rich in catechins and other flavonoids. We then transformed this gene into wild-type Arabidopsis as an over-expression (OX) line 1 and line 2 in ½ MS media to verify its function. In the wild types (WT), the primary root length, number of secondary roots, and total root weight were significantly higher at 24%, 15%, and 53.92%, respectively, compared to the transgenic lines (OX1 and OX2). By contrast, the leaves displayed larger rosettes (21.58%), with higher total leaf weight (32.64%) in the transgenic lines than in the wild type (WT). This result is consistent with DCR mutant *At5g23940* gene in *Arabidopsis thaliana*. Here, anthocyanin content in transgenic lines was also increased (21.65%) as compared to WT. According to the RNA sequencing data, a total of 22 growth regulatory genes and 31 structural genes with TFs (transcription factors) that are correlative with plant growth and anthocyanin accumulation were identified to be differentially expressed in the transgenic lines. It was found that some key genes involved in *IAA* (Auxin) and *GA* (Gibberellin) biosynthesis were downregulated in the transgenic lines, which might be correlated with the phenotype changes in roots. Moreover, the upregulation of plant growth regulation genes, such as UGT73C4 (zeatin), ARR15, GH3.5, ETR2, ERS2, APH4, and SAG113 might be responsible for massive leaf growth. In addition, transgenic lines shown high anthocyanin accumulation due to the upregulation of the (1) 3AT1 and (3) GSTF, particularly, GSTF12 genes in the flavonoid biosynthesis pathway. However, the TFs such as, CCoAMT, bHLH, WRKY, CYP, and other MYBs were also significantly upregulated in transgenic lines, which increased the content of anthocyanins in *A. thaliana* seedlings. In conclusion, a BAHD acyltransferase *(TEA031065)* was identified, which might play a vital role in tea growth and secondary metabolites regulation. This study increases our knowledge concerning the combined functionality of the tea BAHD acyltransferase gene *(TEA031065)*.

## 1. Introduction

The leaf growth and the characteristics of secondary metabolites are critically significant for the quality control and production of plants [1]. Secondary metabolism assists a plant to cope with environmental factors, is sensitive to external changes, and performs a vital role in sustaining plant growth [2]. In addition, these also can promote plant growth due to preventing the disease by redeeming vegetative growth by the increase of root and shoot systems production [3]. The acylation of aliphatic and aromatic both in plants’ secondary metabolites is catalyzed for starter by acyl-CoA-subject acyltransferases family, formerly called the enzymes of BAHD superfamily, which have the donor or acceptor molecules as phenolic compounds [4]. These enzyme families deriving phenolic compounds dominate the secondary compounds acylation in vascular trees [5]. A soluble BAHD acyltransferase family called after the first four biochemically individualized enzymes, BEAT (benzyl alcohol O-acetyltransferase) [6], AHCT (anthocyanin O hydroxycinn amoyl transferase) [7], HCBT (anthranilate N-hydroxy cinnamoyl/benzoyl transferase) [8], and DAT(deacetylvindoline 4-O-acetyltransferase) [9].

The most significant BAHD-ATs studies related to the biosynthesis of CGA have tended to finding homologous enzymes in various plant species [10]. This is also connected with the organization of aromatic compounds by lipid-biosynthesis barriers, such as cutin and suberin. BAHD-ATs from the same clade can acylate a large array of secondary metabolites, and several members of the BAHD family have been shown to be cytosolic [7]. This is why acylation can influence anthocyanin conduction and storage in plant cells [11]. The variety and complexity of secondary metabolites in vascular plants provide these sessile creatures with their capacity for acclimatization [12]. Acylation provides this variation [7]. Phenolic compounds are repeatedly applied as acyl acceptors and/or donors for acylation produced by BAHD-ATs [5]. This is also recognized as essential for improving plant growth, such as shorter rosettes [13], which arbitrate the inhibition or development of post-genital organ amalgamation [14].

Flavonoids with anthocyanins form a massive subfamily of phenylpropanoid metabolites [15]. The enzymes PAL (phenylalanine ammonia-liase), and HQT (hydroxycynnamoyl-quinate transferase), associated with the BAHD family, were indicated to be engaged in the last movement of the secondary metabolites of chlorogenic acid (CGA)biosynthesis in several kinds of plant [16]. The old INRA flint early line F4 in inbred maize has significantly fewer secondary metabolites, but secondary wall related *ZmMYB* with BAHD increases the accumulation rate [17]. However, it remains to be determined whether the BAHD acyltransferase gene family is responsible for making these acylated flavonol glycosides [18]. The transformation of the acyl group and the enzymatic adhesion of anthocyanins through acyl modification is completed exclusively by the BAHD acyltransferases superfamily in the plant [19]. A large clade of acyl-CoA-exploiting enzymes included in various layouts of secondary metabolite alternation is constituted by the family of BAHD acyltransferase [20]. They form a multistep pathway featuring MYB transcription factors [21]. The *VvMYBA* is a definitive regulator of the last step of anthocyanin transport, synthesis, and modification through the upregulation of *Vv3AT*(*ANTHOCYANIN 3-O-GLUCOSIDE-699-O-ACYLTRANSFERASE*), which encodes a protein of BAHD acyltransferase [22].

The overexpression of BAHD acyl acyltransferases (*ABS1/At4g15400*) plays a key role in regulating the plant growth and biosynthesis pathways that produce the dwarf phenotype in Arabidopsis root for the downregulation of GA (Gibberellin) [23]. BAHD acyltransferase mutant genes were largely expressed in internode elongation, indicating that these genes might be engaged in sorghum internodes cell wall to produce p-coumaroylation and feruloylation [24]. Recently, it has been reported that Defective Cuticuler in the Ridges gene, named DCR, a member of the BAHD acyltransferases family [25], has a defective cuticle, with innovating differentiation in plant epidermal cells [26]. This is also connected with the fusion of post-genital organs, the enhancement of plant growth, and the accumulation of secondary metabolism [27] by interactions involving plant organs [13]. In Arabidopsis, DCR also regulates different plant-growth-regulating genes, such as the Zeatin/Cytokinins (*gfc1)* gene, via a two-component signaling pathway [27]. The overexpression of *SLG* (a protein gene in BAHD acyltransferase) in young panicles and lamina joints suggests its role in ruling cell development in those two parts in rice [28].

Tea (*C. sinensis*) is the maximum consumed nonalcoholic quencher and is going increasingly exoteric and generated from buds and young leaves of the tea plant [29,30]. Tea is one of the prime cash crops which contributes effectively adds economic value that was 200 billion USD with a market in 2020 and expected to come to over 318 billion USD by 2025 [31]. China is the prominent producing country and produced approximately 2.79 million metric tonnes in 2019 [32]. It has been also beneficial for human health and protects plant germination due to the high content of theanine, caffeine, terpenoids, and polyphenols (catechins and anthocyanin) which collectively measure the rich flavors [31]. During harvest, the efficiency of tea is enormously involved in the aging of the young tea leaves [32]. Tea has an extensive amount of polyphenols that attracted the attention of more and more for its medicinal value, for example, antiaging, lowering blood sugar, anti-oxidation, lowering blood lipids, anti-atherosclerosis, anti-viral, anti-bacterial, anti-cancer, anti-inflammatory, and treating nerves with pharmacological effects such as degenerative diseases [33]. In addition, rising polyphenols are called flavonoids in fresh leaves in tea that assert various physiological functions and they can be specially and temporarily regulated by plant growth [34]. The secondary metabolites of various plants were catalyzed by the acylation of the BAHD acyltransferases genes family. These genes show the expression in the tapetum of anther cells in the primary stages of leaf, flower, and root development [13,35]. Recently, maximum synthetic gene identification in vivo function becoming applied to systems of heterologous expressions, such as Arabidopsis and tobacco [36].

Despite the recent advances formed by considering the genetic control in the accumulation of secondary metabolites in tea plants, the acylation control genetic mechanism in flavonoids, especially anthocyanin, and the phenotypic modifications in the BAHD acyltransferase (*TEA031065*) gene in this family have not yet been determined. Nevertheless, no gene governing the acylation, especially BAHD acyltransferase (*TEA031065*) gene of anthocyanins, has been characterized in tea plant. We previously analyzed the RNA sequencing of BAHD acyltransferase (*TEA031065*) gene, which was highly expressed in young leaves and stems in tea plants [37]. To evaluate the role of this gene function in tea plants, the *TEA031065* overexpression transformed into a wild-type *A. thaliana.* Next, it was observed that it was significantly expressed in leaves and featured a high accumulation of secondary metabolites, particularly anthocyanin, due to the regulation of the responsible DEGs compared with the control. Moreover, the overexpression of the gene (*TEA031065*) in transgenic Arabidopsis (*A. thaliana*) actuated the maximum flavonoid-involved genes, concluding in the enhancement of anthocyanins in transgenic plants. The RNA sequence analysis of the *TEA031065* gene in all the acylated grounding strongly suggests that it plays a significant role in secondary-metabolite accumulation and plant growth.

## 2. Results

### 2.1. BAHD Acyltransferase Genes’ Expression in Tea Plant (C. sinensis)

As BAHD Acyltransferase genes are important genes involved in secondary metabolism, we screened the gene family from the published tea genome database (TPIA; http://tpia.teaplant.org; accessed on 26 March 2019). A total of 21 BAHD acyltransferase genes were identified in the tea plants. Interestingly, a key gene *(TEA031065)* was identified. Its expression level in young leaves was significantly higher (Figure 1) compared to other BAHD acyltransferase genes at eight representative tissues (apical bud, flower, fruit, young leaf, mature leaf, old leaf, root, and stem), which was consistent with the high quantity of secondary metabolites in tea.

A blast analysis also showed that it favorably encodes a BAHD acyltransferases family member that utilizes CoA thioesters and catalyzes the structure of various sets of secondary metabolites [7].

### 2.2. BAHD Acyltransferase-DCR(TEA031065) Gene Expression in Three Different Tissues in Tea Plant (C. sinensis)

To recognize the genes potentially attached to the phenotype of young leaves, tea samples (roots, stems, and young leaves) from Longjing 43 (LJ43) were subjected to RNA-Seq. analysis [37]. According to the RNA sequencing data, the *TEA031065* gene was highly upregulated in the young leaves and stems in the tea plant (Figure 2). However, the 1-S and 1-R were identified as the stems and roots of newly growing seedlings. In addition, the leaves, stems, and roots of the second- and third-stage seedlings the tea cultivar LJ43 were identified as 2-L, 2-S, 2-R, 3-L, 3-S, and 3-R, respectively. We observed a significant difference in the root samples from the stem and leaf groups. Thus, the eight samples were separated into three groups, called groups R (1-R, 2-R, and 3-R), S (1-S, 2-S, and 3-S), and L (2-L, 3-L). Next, the various patterns of gene expression were explored in group R vs. S, group R vs. L, and group S vs. L. [37]. Similar results were found in the tea cultivar Longjing43 for young leaves, and it is clear that the young leaves and buds are the principal products of tea plants.

### 2.3. Phenotypical Expression and Statistical Analysis of Arabidopsis Root (A. thaliana) among Transgenic Plant and WT

To justify the function of the tea gene, *TEA031065*, we transformed it into wild-type Arabidopsis as an overexpression. A total of fifteen transgenic lines were harvested and, next, two lines were selected for over expression for functional analysis. While growing OX and WT seedlings on vertical agar plates, we observed that the transgenic plants displayed significantly lower expressions of phenotypes in roots compared to the control (Figure 3A). The primary root length was increased by 24% (Figure 3B) and the secondary root length was increased by 15% (Figure 3D) by WT- compared with the OX-line seedlings on day 14. In addition, the total fresh root weight was also 53.92% higher in the WT (Figure 3C) than in the transgenic plants on day 21. Moreover, observing the transgenic plants 14 d after germination, the primary root length (Figure 3B) and lateral root number (Figure 3D) were higher in the wild-type than in the transgenic lines. By contrast, after 21 days of observation, the transgenic plants showed greater root weight than the wild type (Figure 3C). Therefore, there were more lateral root initials with higher primary root length in the wild-type seedlings compared with the transgenic lines.

### 2.4. Phenotypical Expression and Statistical Analysis of Arabidopsis Young Shoots (A. thaliana) among Transgenic Plant and WT

The overexpression (OX) in the transgenic lines showed a clear visible phenotype in the leaves on day 21, as they displayed specious rosettes compared to the WT (wild-type) plants (Figure 4A). The transgenic plants also displayed leaves with fresh weight (Figure 4B) and length (Figure 4C) that were 32.52% and 21.58% higher, respectively, than those of the WT. However, the number of leaves was not significantly different (8.24% only) in the transgenic plants (Figure 4D). Nevertheless, our results indicated that compared to the WT, the *TEA031065*-overexpressing plants has a definite effect on the young leaves’ growth and development in the *A. thaliana* and the tea plants. These results were consistent with the phenotype of the mutant gene *At5g23940* in *A. thaliana* [13].

### 2.5. Anthocyanin Contents in Transgenic Leaves in A. thaliana under ½ MS Media

As BAHD acyltransferase catalyzes secondary metabolites, especially flavonoids, we focused on flavonoid metabolism, including anthocyanin accumulation, in plants, which is classically regulated through transcription factors (TFs). In particular, our attention was focused on the members of this functional class of genes that were differentially regulated among the transgenic lines and wild types in *A. thaliana* under ½ MS media. The 760CRT dual-wavelength, double-beam Prove UV-2500 spectrophotometer was used to determine the anthocyanin accumulation, as shown in Figure 5. We found that the total anthocyanin content significantly varied (45.9% > 35.7% > 18.9%) among the transgenic lines and wild types, and OX1 > OX2 > WT (Figure 5). The anthocyanin contents under WT were remarkably lower than those under the *TEA031065* over-expression (OX) lines in *A. thaliana*. The results provided a meaningful change for flavonoid biosynthesis in the transgenic lines that was significantly greater than in the WT.

### 2.6. Transcriptional Changes in DEGs and the Reference Genome in Transgenic Plants (A. thaliana)

To identify how the tea *TEA031065* (a member of BAHD acyltransferase family) gene affects plant growth and secondary metabolites, we performed RNA sequencing to explore the potential metabolisms. A total of 40.33–44.78 million raw reads were obtained, and the Q20 and Q30 values were larger than 97% and 92%, respectively, displaying significant production and quality. This study detailed nearly 96–98% of the clean read data of the *A. thaliana* genome (Table 1). The RNA-Seq datasets were deposited in the NCBI SRA database under accession number PRJNA874207. After assimilation with the wild type (WT), the global gene transcript affluence was aggravated under *TEA031065* overexpression (Figure 6). The DEG (differentially expressed gene) values obtained in the OX1 vs. WT, OX2 vs. WT, and OX vs. WT groups are reported in Figure 6. Interestingly, compared to that under WT, the total numbers of DEGs under OX1 (overexpression-1); OX2 (overexpression-2) and OX (overexpression) were 695, 779, and 702, respectively. A total of 695 DEGs were recognized between OX1 and WT, with 297 being upregulated and 346 downregulated.

Similarly, 297 and 346 DEGs were upregulated and 482 and 356 were downregulated in the OX2 vs. WT and OX vs. WT groups, respectively (Figure 6). The DEGs Venn diagram showed that 349 genes were commonly found in two groups, as shown in the OX1 vs. WT and OX2 vs. WT comparisons displayed in an Appendix A. The putative function of the DEGs was estimated by the analysis of the KEGG enrichment pathway due to the distribution of the better accession gene functions at the macro level. According to the relative analysis of the three principal groups, OX1 vs. WT and OX2 vs. WT displayed significant similarities in the proportion of the genes, but variations were identified in many subcategories. Most of the GO terms of the DEGs in the OX1 vs. WT and OX2 vs. WT featured the same biological processes, cellular components, and molecular functions (Appendix A). Most of the GO subcategory terms were significant (*p*-value < 0.05) variations in counts within the OX1 vs. WT and OX2 vs. WT. The enrichment analysis of the KEGG pathway was applied to determine the DEGs’ putative functions. The plant-growth-regulation pathways (ath00380, ath04075, ath00350, ath00908, and ath04016), phenylpropanoid biosynthesis (ath00940) and flavonoid biosynthesis (ath00941) were identified across both comparisons. Due to the enrichment of the DEGs within OX1 vs. WT and OX2 vs. WT, the enrichment of the global metabolic pathways for the DEGs was analyzed more closely. In the Appendix A, most of the DEGs were annotated to phenylpropanoid biosynthesis, valine, leucine and isoleucine degradation, phenylalanine metabolism, tryptophan metabolism, flavonoid, plant-hormone signal transduction, the MAPK signaling pathway, plant, the biosynthesis of amino acids, peroxisome, and plant–pathogen interaction.

Among these DEGs, the different biosynthesis pathways of plant-growth regulation, amino acids, and flavonoids in energy metabolism were significantly high. This revealed that the Arabidopsis shoots contained this energy metabolism in the ½ MS media with a large variation. In the enrichment analysis of the KEGG pathway for the OX1 and OX2, 85 and 101 DEGs, respectively, were annotated in the 10 most important pathways (FDR < 0.05). These DEGs included pathways that are shown in Appendix A. The PCA analysis played a role in analyzing their relationships due to the increased investigation of the expression patterns of the gene in three groups Appendix A in plant growth and flavonoid distribution.

### 2.7. Analysis of DEGs (Differentially Expressed Genes) Involved in Plant Growth and Development under ½ MS Media

Surprisingly, the maximum structural DEGs in plant growth and development were significantly upregulated in the OX lines, except for a few genes, to varying degrees (Figure 7). Our results showed that tryptophan metabolism, diterpenoid biosynthesis, the MAPK signaling pathway, plant-hormone signal transduction, and the zeatin biosynthesis pathways were the most highly produced plant-growth and developmental pathways in the *A. thaliana*. Most of the enzymatic genes in these pathways, such as ALDH7B4, NIT2, CAT, ETR, AHP4, and GH3.5, were upregulated in tryptophan metabolism; the UGT73C4 in zeatin biosynthesis pathways and the SAG113, ERF13, and ERS in the MAPK signaling pathway were significantly induced by the overexpressed (OX) transgenic lines (Figure 7).

By contrast, IAA, SAUR in tryptophan metabolism, ARR15, PYL6 in the plant-hormone transduction signal, GA3OX1, and GA20OX1 in the diterpenoid biosynthesis pathways were downregulated in the transgenic (*TEA031065*) lines. Compared to the transgenic lines, the WT inhibited the plant-growth-metabolism pathways by downregulating the IAA, ARR15, and GA in the *A. thaliana* (Figure 7). In addition, the IAA and SAUR genes in the tryptophan-metabolism pathway showed comparatively high differences regarding their condensation across all the transgenic lines. The production rate of the features of plant growth, such as the primary root length and the number of lateral roots, were largely changed across the *TEA031065* treatments. These compounds provided a greater accumulation level in the WT than in the transgenic lines.

Moreover, the candidate genes functioning naturally had prominent expression levels in the transgenic lines when their volume of bio product was extremely high, and 22 potentially key functional genes were acquired. These genes included ALDH7B4 (1), NIT2 (1), CAT (2), UGT73C4 (1), WRKY (1), SAG113 (1), ERF1B (1), NDK1 (1), ERS2 (2), SPCH (1), AHP4 (1), GH3.5 (1), SAUR (1), IAA (2), ARR15 (1), PYL6 (1), and GA (2) (Figure 7). The accumulation of plant growth and production were regulated by the DEGs, which were found to perform an important role in this regard. According to the RNA-Seq. data, the 21 DEGs (differentially expressed genes) were categorized from six families from the OX1 vs. WT and OX2 vs. WT groups for growth and development in *A. thaliana.*

These results indicated that it could lead to a distinct hormone-signaling regulation, which may perform an essential role in the growth and development process in tea plants.

### 2.8. Analysis of Candidate Genes with Anthocyanin Production in A. thaliana

The 3 GSTF (GSTF3, GSTF7, GSTF12), 1(3AT), and 2 UGT (UGT73 and UGT75) genes with the regulatory genes, 6 CYPs, 3 WRKYs (WRKY33, WRKY53, and WRKY75) 3 MYBs (MYB32, MYB112, and MYB51), CCoAMT, and bHLH in the flavonoid biosynthesis pathway were upregulated in the transgenic lines but downregulated in the WT (Figure 8), resulting in the increased synthesis of anthocyanins under *TEA031065*-overexpressed (OX) transgenic lines (Figure 5). The young shoots of the *A. thaliana* showed increased anthocyanin synthesis due to the upregulated expression of the DEGs under the effects of the over-expressed (OX) BAHD (*TEA031065*) acyltransferase gene, whereas the anthocyanin synthesis decreased in the wild types (WTs).

### 2.9. Analysis of DEGs (Differentially Expressed Genes) Involved in Flavonoid Biosynthesis under ½ MS Media

According to the RNA sequencing data, most of the DEGs were significantly upregulated in the transgenic lines (*TEA031065* overexpression) in the *A. thaliana*, except for few genes, to asymmetrical degrees (Figure 8). Our results suggest that the most frequent metabolic outcomes, such as anthocyanins and flavonoids, occurred through the phenylpropanoid and flavonoid biosynthesis pathways. In these pathways, the maximum enzymatic genes, such as BGLU, CAD, 3AT1, CCoAMT, UGT, and GSTF, were significantly induced by the *TEA031065*-overexpression (OX) lines (Figure 8). In the WT, the highest enzymatic gene expression level in the pathway significantly decreased (Figure 8). By contrast, GSTF12 and UGT were downregulated in the WT. In comparison with the transgenic lines, the WT suppressed the anthocyanin metabolism by downregulating the GSTF, 3AT, and CCoAMT in the Arabidopsis plant (Figure 5). Moreover, the transgenic lines showed the maximum functional genes with high-level expression. The bio-product contents were significantly higher in the transgenic lines because 13 potentially essential gene functions were obtained. These genes included PAL (1), 4CL (1), CHS (1), CHI3 (1), LDOX (1), CCoAMT (1), CCR3 (1), 3AT1 (1), GST (2), UGT (2), and GSTF12 (1) (Figure 5 and Figure 8).

TFs are known as key factors in regulating plant growth and production. According to the RNA sequencing data, the CYP, WRKY, bHLH, and MYB TFs played a vital role in accumulating the anthocyanin synthesis in this study. In addition, the 21 DETFs (differentially expressed transcription factors) were recognized from eight families in the OX1 vs. WT and OX2 vs. WT groups (Figure 5 and Figure 8). These DETFs represented the most common TF families, which were PER (8, 25.80%) and CYP (6, 19.35%), followed by BGLU (5, 16%), WRKY (4, 12.90%), MYB (3, 9.68%), CAD (2, 6.45%), bHLH (1, 3.22%), PCL1 (1, 3.22%), H1H9.1 (1, 3.22%), and HPD (1, 3.22%). Surprisingly, the MYB and bHLH were highly expressed in the transgenic lines and had a balance with the expression of the linked key DEGs supported in flavonoid biosynthesis (Figure 8). They controlled the expression of the regulatory genes and performed an essential role in anthocyanin regulation. Among them, one TF similar to WRKY44 might play key roles in flavonoid regulation [37]. In addition, GSTF12 *(AT5G17220)* was highly similar to CsGSTF1 *(Cha08g011300)* [38,39], a principal regulator of anthocyanin production. Overall, many DEGs displayed that the individual patterns of expression in the WT and transgenic lines were related to flavonoids and lignin metabolism. Interestingly, the 31 DEGs associated with flavonoid biosynthesis were significantly upregulated in the transgenic lines in the *A. thaliana.* Further research on these TFs would increase our knowledge of the regulatory system for flavonoids and anthocyanin biosynthesis (Figure 8).

## 3. Discussion

### 3.1. TEA031065 Expression Pattern of BAHD Acyltransferase Genes Family in Various Tissues of Tea (C. sinensis) Plants

A large group of BAHD genes accessible from sequenced plant genomes offers an advantage in expanding the exploration of the preservation strategy in this family with two close functional domains, HXXXD and DFGWG [40]. According to the report of the phylogenetic tree, it is extensively compatible with a previous study [40], which categorized biochemically individualized BAHD acyltransferases into different groups. In addition, an acyltransferase-like gene (*LaAT*: *Lupinus angustifolius acyltransferase)* showed a potential homology with the BAHD acyltransferases in the *A. thaliana*. The expression was highest in the young leaves, but barely detectable in the other organs of the bitter cultivar plant of *Lupinus angustifolius* [41]. The encoded *CHAT* (acetyl CoA: (Z)-3-hexen-1-ol acetyltransferase) gene was also highly expressed in the leaves and stems in *A. thaliana* [42]. Our study explored the expression of twenty-one BAHD acyltransferase genes from eight different tissues in tea (*C. sinensis*): apical buds, flowers, fruits, young leaves, mature leaves, old leaves, roots, and stems (Figure 1). Among them, the *TEA031065* gene expression in BAHD acyltransferase was significantly increased in the young leaves and stems, which was consistent with the RNA seq. analysis data (Figure 2) in *Longjing 43* (LJ43), a green-tea cultivar (*C. sinensis*) [37]. Our results suggested that the newly discovered motifs, especially the gene expression of BAHD acyltransferase in different tissues, particularly *TEA031065*, should simplify the future functional research of substrate and donor certainty within BAHD enzymes in tea.

### 3.2. Phenotyping Expression of TEA031065 in A. thaliana

Since the brittleness of young tea leaves and buds can improve the product quality of tea plants [43], we then transformed the *TEA031065* gene into the Arabidopsis wild type as an overexpression in ½ MS media to verify its function (Figure 3 and Figure 4). Previously, DCR was known as a prime gene whose expression is significantly responsible for cutin metabolism [13]. Many racial phenotypes were closely linked with a defective cuticle due to the mutation of DCR; this was shown in the *A. thaliana* for the T-DNA *gfc1* mutant gene [27]. A previous study reported that the mutant of *lateral root development2* was subject to the osmotic restraint in the formation of lateral root due to the mutation of the candidate gene, *Long Chain Acyl-CoA Synthetase2*, to cutin biosynthesis [44]. The DCR expression mainly affects the vegetative and reproductive organs in epidermis cells, but the development of lateral-root primordia and root cap resulted in the high expression of this gene [45]. Our research showed that the *TEA031065*-overexpressed (OX) transgenic lines reduced the primary root length, along with the lateral root number and growth weight, compared with the WT in the *A. thaliana* (Figure 3). The reduction in the additional root branching and formation of root length in the OX lines featured the contribution of the *TEA031065* overexpression in the lateral root initiation in the *A. thaliana* [13]. In this study, *TEA031065* gene overexpression (OX) in newly emerging leaves and shoot elongation with leaf numbers revealed that it is also included in the epidermis cell wall in vegetative organs (Figure 3). Furthermore, the leaf weight was also used to describe the developmental phenotype in the transgenic lines compared to the wild types (Figure 4). The results were consistent with the phenotype change of the *At5g23940* in Arabidopsis DCR mutant genes [13]. The importance of these shapes is clear, and *TEA031065* might play a role similar to DCR in the plant growth and developmental system, either by enhancing the cell surface to assist the escape of flavor molecules or by stimulating the reflex from vegetative cells for attractive phenotypes [46].

### 3.3. TEA031065 Might Play a Role as a Regulator for Plant Growth and Development Gene in Tea (C. sinensis) Plant

The seedlings containing *gfc1* (zeatin) from the BAHD acyltransferase mutant gene revealed the higher marking of the cell-cycle reporter *pCYCB1: GUS* increasing the primary root length with fresh root weight under natural growth conditions [27]. This revealed the DCR’s functional role in influencing the cell division and segregation in *A. thaliana* [27]. Our results reveal that the *TEA031065* gene encoding acyltransferases that incite the exit of plant-growth-regulation genes through different plant-growth-signaling biosynthetic pathways, such as zeatin and tryptophan metabolism, diterpenoid biosynthesis pathways, MAPK signaling biosynthesis pathways, etc. (Figure 7). In the *TEA031065*-overexpressed lines, AHP4 and UGT73C4, the genes for Zeatin metabolism were significantly upregulated compared to the wild types, whereas ARR15 was highly downregulated in *A. thaliana*, which might have increased the leaf initiation and leaf growth (Figure 3, Figure 4 and Figure 7). In addition, based on the RNA sequencing analysis, many plant-growth-regulatory genes, such as CH3.5, ETR2, and ERS2, etc., which are involved in different plant-growth-hormone-signaling pathways, were highly upregulated in the overexpressed transgenic lines (Figure 7). In the cytokinins (CK) mutant DCR gene *(gfc1)*, the maximum type-A ARR15 (Arabidopsis Response Regulator) genes’ expression levels were upregulated, which reduced the adventitious root growth under MS medium [27]. Plant hormones are the principal signal molecules regularizing plant growth and development and Gibberellins *(GAs)* are the first class of plant hormones that improve plant growth [47]. The *IAA* (Auxin) accumulation was highly reduced in the young parts of the plant organs by overexpressing CBF2 (C-repeat binding factors) and CBF3 compared with the WT in the *A. thaliana* [48]. As in the *dcr-1*, the branching of the root was also influenced in the *bdg* [49] and the *dso* [50] genes in the *A. thaliana*, but we did not observe any important changes in the composition of the suberin polyester in the roots of the *TEA031065* overexpression compared with the WT roots. In this study, the primary root length, secondary root numbers, and root weight were significantly higher, by 24%, 15%, and 53.92%, respectively, in the wild type by upregulating the IAA (IAA5 and IAA6) and GA (GA3OX1 and GA20OX1) (Figure 3 and Figure 7). Interestingly, the expression of the UGT73C4 genes in zeatin (cytokinin) metabolism was highly upregulated in the transgenic lines and balanced the structural DEGs expression influencing the cell development and shoot initiation. By contrast, the ARR15, IAA, and GA expression were downregulated in the transgenic lines, which might have decreased the primary root length and lateral root initiation compared to the WT (Figure 3). In addition, two GA (Gibberellin) genes (GA3OX1 and GA20OX1) were upregulated in the WT, which increased the root growth in the *A. thaliana* (Figure 3 and Figure 4). Moreover, we found that most of the plant-growth regulatory gene members were highly upregulated in the OX lines (Figure 7). In particular, the UGT73C4 (AT2G36770), APH4A (AT3G16360) GH3.5 (AT4G27260), IAA (AT1G52830 and AT1G15580), and GA (AT1G15550 and AT4G25420) were significantly and positively correlated with plant growth and development (Figure 3, Figure 4 and Figure 7).

### 3.4. The Importance of TEA031065 in Tea (C. sinensis) Secondary Metabolites

In various plants, the growth process was involved in secondary metabolites called phenolic compounds. In addition, a broad phenolic product frequently comprises an acylation step led by the biosynthetic pathways. This acylation also supports the transportation and storage of anthocyanin in plant cells [5]. The CQA (caffeoylquinic acid) content in *Nicotiana benthamiana* was influenced not only by BAHD acyltransferases but also by the structural genes of the phenylpropanoid pathway (PAL, 4CL, C4H) and the regulatory genes (such as MYB, bHLH, and other transcription factors) [51]. Our study suggests that the maximum DEGs in different metabolic pathways were more highly upregulated in the transgenic lines than the WT in the Arabidopsis leaves (Figure 5, Figure 6 and Figure 8, and Table 1). This result suggests that the TFs with DEGs under different metabolic pathways were more highly increased in the *TEA031065* transgenic lines compared to the WT (Figure 7). In the Arabidopsis plants, the transgenic lines also provided more secondary metabolites, particularly anthocyanin, than the wild types (Figure 5).

In this study, *TEA031065* increased the anthocyanins content (Figure 5) in the leaf vacuole of the *A. thaliana* by upregulating the structural genes, such as, GSTF, AtGSTF12, UGT (UGT75D), and TFs (bHLHs and MYBs) in the transgenic plants (Figure 8).

Previously, it was observed that the outcome of UGT75C1 (*At4g14090*) catalyzes the essential step of 5-O-glycosylation for synthesizing the leading anthocyanin compounds in *A. thaliana* by influencing the BAHD acyltransferase melonyl CoA gene [52].

Our results revealed that a total of 47 structural and TF genes were influenced in their flavonoid production. Moreover, additional DEGs and TFs (Figure 8) were upregulated in OX1 vs. WT and OX2 vs. WT, which is why the total anthocyanin accumulation (Figure 5) was higher in the transgenic lines than in the WT. The expression of the structural genes, such as GSTF (4), UGT (2), and 3AT (1) included in the pathways of flavonoid biosynthesis, was significantly upregulated in the *TEA031065*-overexpression lines (Figure 8). In the grape vines, *Vv3AT (ANTHOCYANIN3-O-GLUCOSIDE-699-OACYLTRANSFERASE)* belongs to a BAHD protein clade responsible for a range of functions, which is capable of producing and modifying the acylated anthocyanins commonly found in grape berries [22]. Moreover, we found that the maximum TF members, together with the PER (3/6), CYP (6/6), MYB (3), WRKY (4), and bHLH (1) genes, were highly upregulated in the *TEA031065*-overexpressed (OX) lines (Figure 8). In particular, *CsGSTF12 (Cha08g011300)*, was highly and positively connected with anthocyanin accumulation (Figure 5 and Figure 8), which is consistent with our previous studies [38,39]. The pathways of phenylpropanoid biosynthesis, the structural DEGs, such as CHI, CHS, DFR, CYP98A, FLS, F3H, and ANS, were placed in the middle and last steps, where the anthocyanin content was naturally correlated with these late biosynthetic genes in apple skin [53,54].

However, the secondary metabolites, particularly anthocyanin accumulations, were significantly enhanced in the *TEA031065*-overexpression lines compared to the wild types in the *A. thaliana* in ½ MS medium (Figure 5). In addition, anthocyanin synthesis had a positive regulatory outcome by the auxin (IAA) and abscisic acid (ABA), whereas Gibberellin (*GA*) had opposite effect on the mutant bud sport apple (*Malus domestica Borkh*.) [55]. The key genes with TFs, such as MYB, bHLH, PER, BGLU, and CCoAMT, related to plant-hormone signal transduction, phenylpropanoid and terpenoid biosynthesis, and flavonoid biosynthesis pathways, immediately influenced the anthocyanin accumulation (Figure 8). The biosynthesis of the anthocyanin was promoted by the functionally characterized cMYB7, MYB, and DcMYB6 genes in both carrot and Arabidopsis due to stimulation of acylation [56]. However, we concluded that *TEA031065* might be more useful for plant growth and development with the accumulation of secondary metabolites, especially anthocyanin, than the WT by upregulating the gene expression in plant-growth-hormone signaling and flavonoid-biosynthesis pathways. By contrast, WT could provide the maximum root growth in tea plants. Following the observations, various conditions, such as plant species, time, and season, can affect these results by changing the gene expression in plant-growth signaling pathways and flavonoid-biosynthesis pathways.

## 4. Materials and Methods

### 4.1. Generation of Transgenic Arabidopsis Seedlings Overexpressing BAHD Acyltransferase (TEA031065)

The CDS of *TEA031065* was enhanced by PCR from transcriptome cDNA of the LJ43 tea plant, and the distinctive primers were TEA031065-SacI-F: 5′ CAGTCCCTGTCATCGCACTT 3′ and TEA031065-SalI-R: 5′ TTCTGTATCAACCCGGCACC 3′. The OX (overexpression) plasmid *TEA031065* was initiated into *Agrobacterium tumefaciens* and transformed into *A. thaliana* (WT, Col-0) by applying the floral dipping action [57,58]. The T1 and T2 propagations of transgenic plants were screened applying 25 μg mL^−1^ hygromycin on 1/2 Murashige and Skoog medium (MS medium), and the T3 homozygous strain lines, after verification by overexpression analysis (2^−ΔCt^ method on the basis of expression in reference β-6-tubulin in Arabidopsis) were applied for further analysis. The Arabidopsis plants were implanted in 1/2 MS plate medium raised in plant incubators at 120 μmol m^−2^ s^−1^ light intensity under 16/8 h light/dark and 70% relative humidity, at a temperature of 22 °C/18 °C. This method followed that adopted in [59].

### 4.2. Plant Growth Conditions and Over-Expression Lines

To observe the various phenotype characteristics with molecular mechanism of *A. thaliana*, this research was performed in the growth room at the Institute of Tea Research, CAAS, and Hangzhou. Using a wild-type *A. thaliana* T3 population segregating for functional analysis, the specimens were grown in the climate room at 22.8 °C, 70% relative humidity, and 16 h/8 h light/dark cycle under WL (Control/CK), OX1 (overexpression of the Arabidopsis BAHD acyltransferase gene line 1 and OX2 (over expression of the Arabidopsis BAHD acyltransferase gene line 2).

For collecting samples (leaf and root), seeds were surface-sterilized in 75% ethanol alcohol for 10 min and then rinsed by distilled water four times. The seeds were vernalized, then imbibe for 2 to 3 d at 4 °C fridge. For all root and leaf branching explorations, 5 to 10 seeds were implanted on Petri dishes holding ½ MS media, as described below and in the study. Plates were grappled with parafilm and placed vertically for growing roots properly on the media surface. Next, the plates were placed into the growth chamber in a controlled environment at 22.8 °C temperature with 16 h/8 h light–/dark cycles. Agar drops with ½ MS media compositions, as noted in the study, comprised 1.2% agar. The agar was cooled until it was introduced to solidify gradually. Every plate contained 100 µL liquid ½ MS media for growing leaves and secondary roots. After 21 days, we collected the Arabidopsis young leaves for transcriptome and metabolome analyses, which were completed with three biological replicates in every treatment.

### 4.3. Anthocyanin Analysis

The accumulation of total anthocyanin was determined with a spectrophotometer named dual-wavelength double-beam Prove UV-2500, 760CRT, using the modified method of [60]. The leaf sample of Arabidopsis (0.2 g) was mixed at 70 °C for two h with 1.5 mL 2% HCl methanol in a water bath with alternated shaking (10 son vortex mixer). The extract was filtered, and its liquid was determined at 530, 652, and 620 nm. The measurement of anthocyanin quantity was based on the formula in previous studies [40,42]. All data are presented as the mean ± SD (*n* = 3). Significance was determined via one-way analysis of variance, and for significant differences within the treatments, the LSD (least significant difference) *t*-test was applied (*p* < 0.05).

### 4.4. Analysis of RNA Sequencing

Here, total RNAs of Arabidopsis leaves samples were extracted by an RNAprep purePlant Kit (Tiangen, Beijing, China). RNA quality was confirmed by 1% agarose gel and the concentration was measured through NanoDrop 2000 (Thermo Scientific, Beijing, China). According to the method described in a previous study, a total of 9 samples were chosen for Illumina Nova Seq. 6000 sequencing [61]. By following the process to the Illumina Nova Seq. 2500 instrument (Illumina, San Diego, CA, USA), applying a customer sequencing service (Beijing Novogene Co., Ltd., Beijing, China), the total samples of Arabidopsis plant were sequenced (125-base-pair paired-end reads). The acquired data sequence was BLAST-investigated in opposition to NCBI non-redundant (NR) protein, SwissProt, KOG (Eukaryotic Orthologue Groups), and KEGG (Kyoto Encyclopaedia of Genes and Genomes) databases, applying an E-value cut-off of 10^−5^. The Blast2go software was applied for functional annotation by GO (Gene Ontology) terms (www.geneontology.org accessed on 21 August 2022). According to the Arabidopsis genome, the sequencing reads were grouped using HISAT2, v. 2.0.4 (Hierarchical Indexing to Splice Array of Transcripts) [62,63].

### 4.5. Identification of DEGs

The Arabidopsis leaf samples’ gene-expression levels were normalized and determined following FPKM (fragments per kilo base of transcript per million reads) values. The PCA (principal component analysis) was compared across the three groups (group OX1 vs. WL, group OX2 vs. WL, and group OX vs. WT). DEGs were recognized by applying Cuffdiff with blind-dispersion procedure and an FDR ≤ 0.01 false discovery rate [64]. The key DEGs with a *p* value ≤ 0.05 and a minimum 2-fold change in expression of diagonal classes were kept. Moreover, blastx was employed to annotate DEGs enrichment pathway analysis of the KEGG (Kyoto Encyclopaedia of Genes and Genomes) [65].

### 4.6. Data Analysis

To determine the candidate genes following FPKM values with the accumulation of anthocyanin, SPSS (Statistical Package for the Social Sciences) Statistics 17.0 was applied for bivariate correlation analysis. Here, we applied Pearson correlation analysis, where *p* values < 0.05 revealed the effective values, and *p* values < 0.01 indicated largely significant values. Letters indicate statistical differences (*p* ≤ 0.05) according to an LSD test.

## 5. Conclusions

In this study, the identification, purification, and characterization of tea *(C. sinensis)* gene *TEA031065* (a member of BAHD acyltransferase family) over expression in wild-type *A. thaliana* were carried out for the first time. The present study highlights the existence of a cytosolic BAHD acyltransferase gene in plants and provides an insight into the molecular and physiological function of the *TEA031065* gene in tea *(C. sinensis)* plant. In leaves, transgenic plant provides large phenotype, most probably by the upregulation of different types of plant hormone signaling pathways genes, including zeatin, tryptophan, Plant hormone signal transduction MAPK and diterpenoid biosynthesis pathways. In roots, wild type shows extensive lateral root formation by the upregulation of IAA and GA biosynthesis genes. Here, we show that *TEA031065* do not only play a role in plant development but are also active in secondary metabolites (anthocyanins) accumulation. The results suggest that GSTF3 (AT2G02930), GSTF7 (AT1G02920), GSTF12 (AT5G17220), and 3AT1 (AT1G03940) genes were highly and positively correlated with anthocyanin accumulation that may be effective and helpful in exploring the complicated metabolism of the transgenic plant and different biosynthesis in tea plant during the growing period. The GSTF12 (AT5G17220) gene may be responsible for increasing anthocyanin accumulation, as it was induced by BAHD acyltransferase *(TEA031065)*. Our study provides a potential consciousness based on the evolutionary relationship of BAHD members for understanding and genetic function, especially *TEA031065* coated in plant growth, development, and flavonoid metabolism. As tea plants are extremely enriched with various types of secondary metabolites and particularly of interest for making tea, future studies on DEGs and of TFs using the tea plant as a model could be enhanced with data on metabolism and development and in the context of different stresses.

## Figures and Tables

**Figure 1 plants-11-02483-f001:**
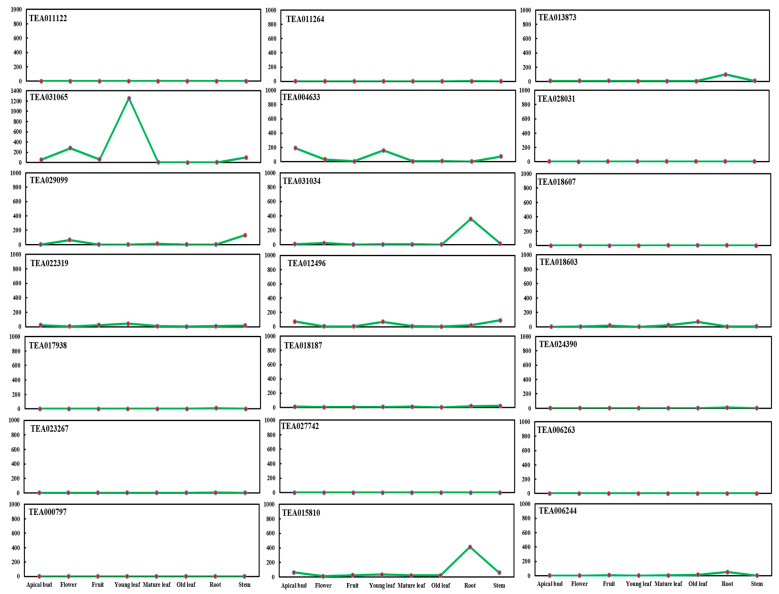
Expression of 21 BAHD Acyltransferase genes in tea (*C. sinensis*). Expression level of tea plant 21 BAHD acyltransferase genes at eight representative tissues (apical bud, flower, fruit, young leaf, mature leaf, old leaf, root, stem). The means of RNA-seq. analysis data are the ± SD (*n* = 3) *p* ˂ 0.05.

**Figure 2 plants-11-02483-f002:**
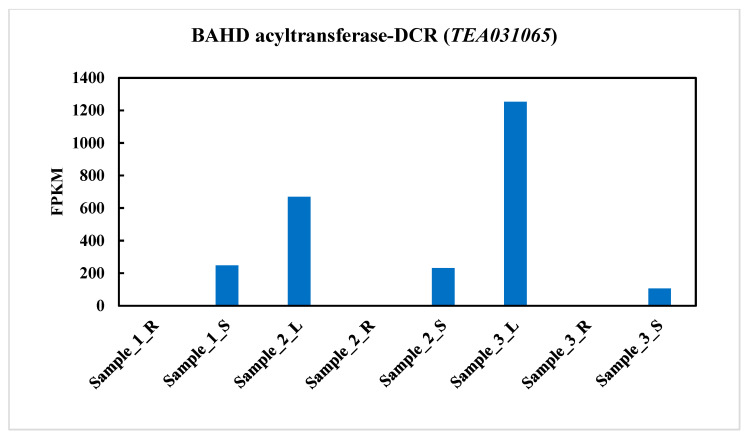
Expression level of BAHD acyltransferase (*TEA031065*) gene in three different tissues (root, stem, and leaf) in tea plant (*C. sinensis*). The eight samples were divided into three groups: 1-R, newly growing root; 1-S, newly growing shoot; 2-L, second-stage leaf; 2-R, second-stage root; 2-S, second-stage stem; 3-L, third-stage leaf; 3-R, third-stage root; 3-S, third-stage stem. The means of RNA-seq. analysis data are the ± SD (*n* = 1) *p* ˂ 0.05.

**Figure 3 plants-11-02483-f003:**
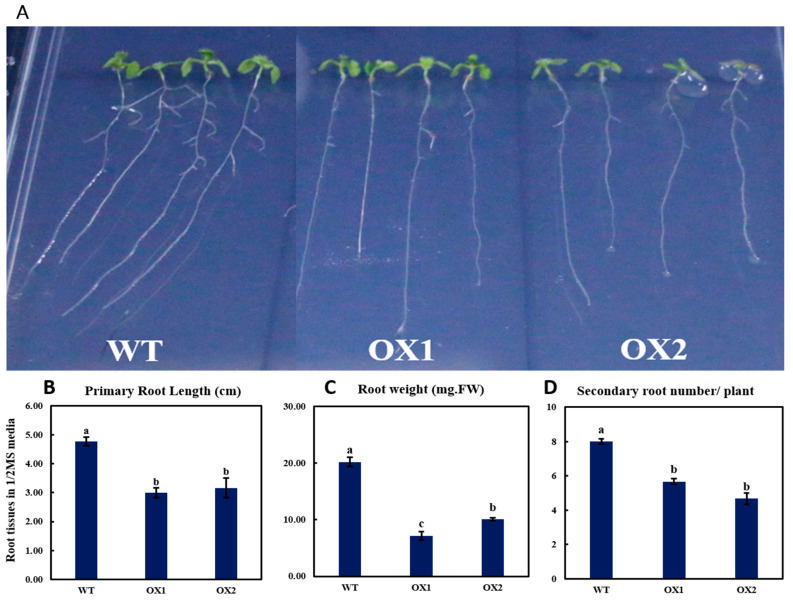
Light-microscopy images of different *TEA031065*-over-expression allele phenotypes for roots and their statistical growth rates in transgenic plants (*A. thaliana*). (**A**) Phenotypes of 21-day-old OX-1 and OX-2 versus wild type; (**B**) primary root length; (**C**) root weight; and (**D**) secondary root number/plant of control (CK)/wild-type and transgenic (lines 1 and 2 gene overexpression) plants for 14 days. The LSD analysis shown the significant differences among the different treatment by the lowercase letter a_>_b_>_c. The values imply means of three biological replicates ± SD. *p* ˂ 0.05.

**Figure 4 plants-11-02483-f004:**
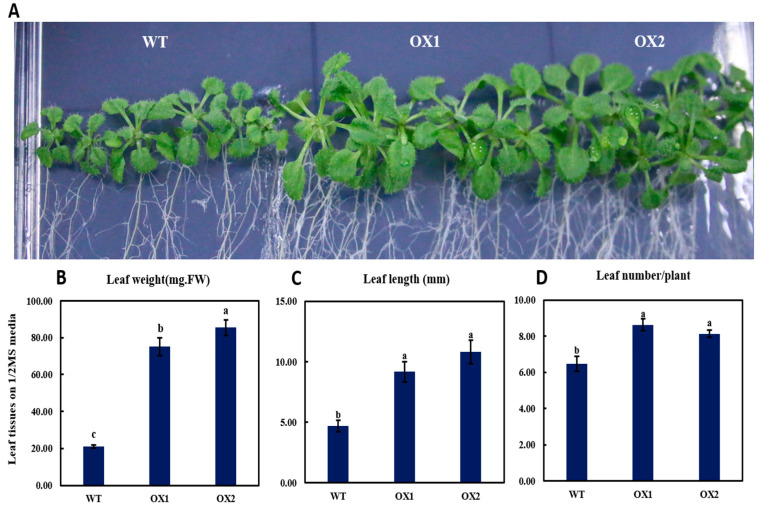
Light-microscopy images of different *TEA031065*-over-expression line phenotypes for leaves and their statistical growth-rate capacity in *A. thaliana*. (**A**) Phenotypes of 21-days old OX-1 and OX-2 versus wild-type leaves; (**B**) leaf fresh weight; (**C**) leaf length; and (**D**) leaf number/plant of control (CK)/wild-type and transgenic (overexpression) lines 1 and 2 gene plants for 21 days. The LSD analysis shown the significant differences among the different treatment by the lowercase letter a_>_b_>_c. The values imply means of three biological replicates ± SD. *p* ˂ 0.05.

**Figure 5 plants-11-02483-f005:**
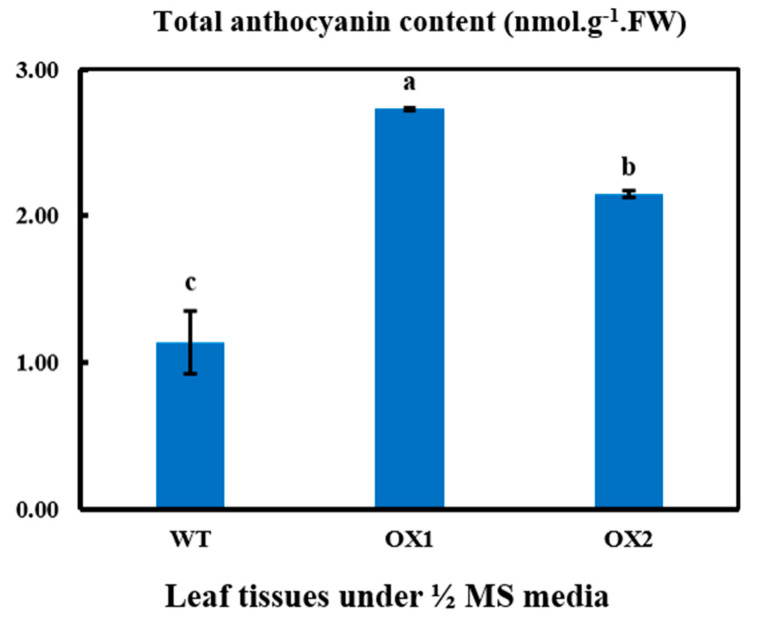
Total Anthocyanin analysis in *A. thaliana* under ½ MS media. Anthocyanin accumulations of newly grown Arabidopsis transgenic leaves in OX (line 1 and line 2) and WT (wild type) by 760CRT dual-wavelength, double-beam Prove UV-2500 spectrophotometer (mean ± standard deviation, *n* = 3). Means showing effective difference (*p* < 0.05) are labeled with different letters based on one-way ANOVA.

**Figure 6 plants-11-02483-f006:**
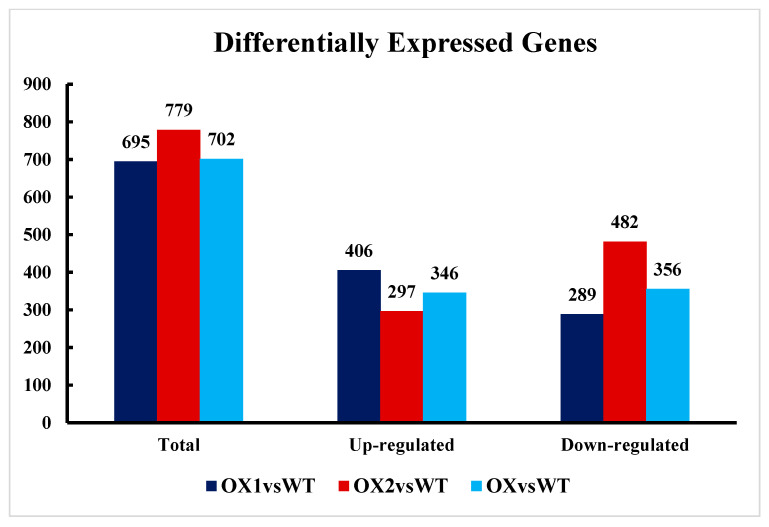
DEGs (differentially expressed genes) in OX1 vs. WT, OX2 vs. WT, and OX vs. WT groups in *A. thaliana*. The means of RNA-seq. analysis data are the ± SD (*n* = 3) and means showing effective difference (*p* < 0.05).

**Figure 7 plants-11-02483-f007:**
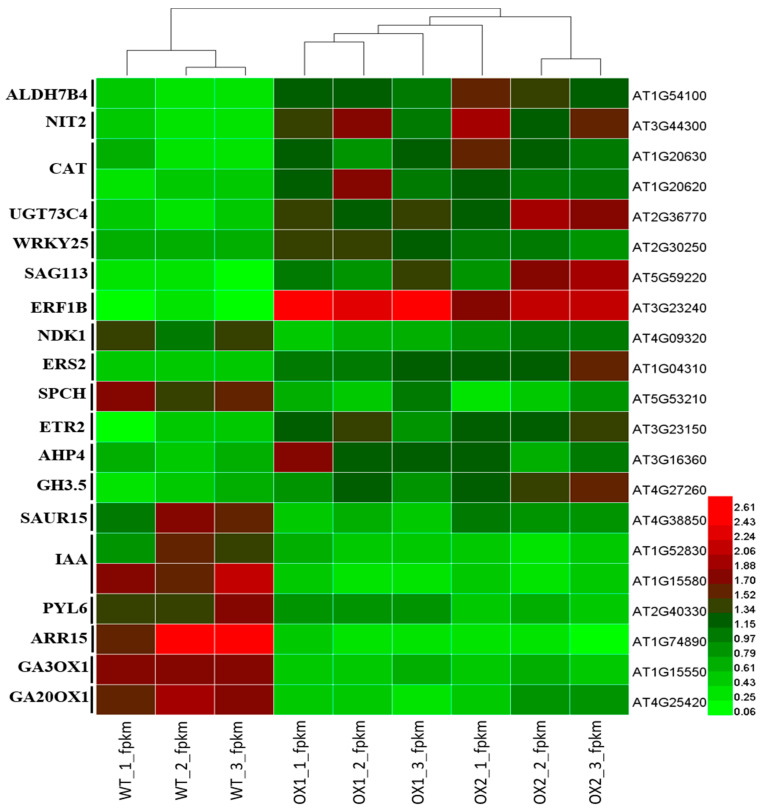
Expression levels of potential DEGs for plant-growth regulation in transgenic plant (*A. thaliana*). The heat map was originated from the log2-fold change (log2FC) and mean value calculated from three replicates of RNA-Seq. data. The color shows the fold change of DEGs under OX1, OX2, and WT, displayed on the right side. The means of RNA-seq. analysis data are the ± SD (*n* = 3) and means showing effective difference (*p* < 0.05).

**Figure 8 plants-11-02483-f008:**
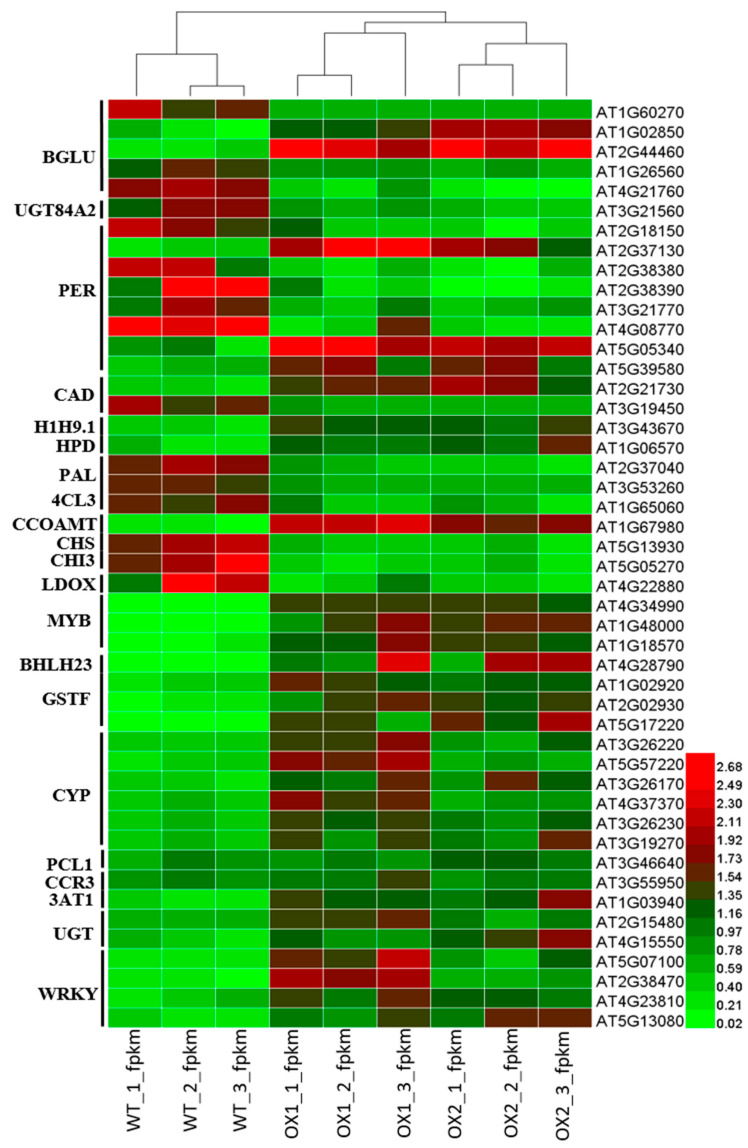
Expression levels of potential DEGs for flavonoid production in transgenic plant (*A. thaliana*). The heat map was created through the log2-fold change (log2FC) and mean value calculated from three replicates of RNA-Seq. data. The color shows the fold change of DEGs under OX1, OX2, and WT, displayed on the right side. The means of RNA-seq. analysis data are the ± SD (*n* = 3) and means showing effective difference (*p* < 0.05).

**Table 1 plants-11-02483-t001:** The RNA-Seq. Data sets’ Summary of Transgenic plant (*TEA031065*) gene in *A. thaliana*.

Sample	Number of Reads (Million)	Total Base (Gb)	Q20%	Q30 (%)	Total Mapped (Million)	Mapping Rate (%)
WT_1	40.33	6.28	97.24	92.12	39.39	97.67
WT_2	43.43	6.74	97.07	91.76	42.37	97.56
WT_3	43.41	6.88	97.24	92.11	42.34	97.53
OX1_1	43.83	6.86	97.3	92.26	42.82	97.68
OX1_2	44.78	7.11	97.33	92.33	43.67	97.50
OX1_3	40.83	6.52	97.2	92.02	39.76	97.38
OX2_1	43.21	6.87	97.25	92.14	41.97	97.13
OX2_2	42.52	6.75	97.32	92.29	41.42	97.41
OX2_3	42.23	6.88	97.42	92.6	40.95	96.96

WT, Wild type; OX, over expression of *TEA031065* gene in transgenic lines.

## Data Availability

Not applicable.

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
