# Peer review of "Identification of a BAHD Acyltransferase Gene Involved in Plant Growth and Secondary Metabolism in Tea Plants"

_plants, 2022, doi:10.3390/plants11192483_

Round 1
Reviewer 1 Report
The manuscript has scientific merit, but English is poor, and needs extensive English language corrections. The authors are advised to get the manuscript rewritten and submit again.
Reviewer 2 Report
This article present Identification of a BAHD Acyltransferase gene involved in plant growth and secondary metabolism in tea plants. Before recommending this article for publication, there are some shortcomings for that should be resolve.
General comments
Overall, the study is well designed and presented in a good way, but mostly the literature is not cited. Grammatical and typos must be revised
Abstract
Remove the “was the maximum”
Greater is not appropriate word “greater rosettes”
Methods results and conclusion must be in sequence in abstract section.
Introduction
The introduction section is started with direct mechanism of acylation which is not bad but to make more informative the authors should add the significance and importance of secondary metabolites and its role in growth, subsequently providing background for the discussion of line 43-50.
Line 54-57 references are missing.
Add economic and medicinal importance of the studied species.
Results discussion and methods are well presented but mostly literature is not cited.
Also add reference in section 4.4 the following reference could be cited.
https://doi.org/10.1007/s10725-021-00785-7,
In section 4.5 the following study could be cited https://doi.org/10.3389/fgene.2021.635043,
Conclusion
Conclusion is well justified.
Reviewer 3 Report
1. "several members of the BAHD family have been shown to be cytosolic" Please provide citation for this statement
2. SD values were not visible in Figure 2 and 6.
3. In the material and methods section author should explain what are the statistical test used in this study by creating new subsection for this
4. Please provide the p values in text of the results section
5. Please provide suggestion for future study based on current findings in the conclusion section
Reviewer 4 Report
The manuscript needs MAJOR REVISION before acceptance for publication. The suggestions/ comments are given below:
(1) The manuscript needs thorough editing for English language
(2) Line No. 12: 'subject' what does it refers to?
(3) Line No. 13: delete 'a' synthesis
(4) Line No. 14: replace 'significant' with 'important'
(5) Line No. 48: rewrite as 'first'
(6) Line No. 96: Italicise 'Camellia sinensis'
(7) Line No. 111: Rewrite as 'phenotypic'
(8) Line No. 121: Rewrite as 'A. thaliana' as it is already presented in full form at Line No. 117. Check for the same throughout the text.
(9) Introduction: Rewrite it briefly with significant points as the Introduction is too lengthy and no flow of the content
(10) Line No. 126: Rewrite as 'C. sinensis' as it is already presented in full form at Line No. 96. Check for the same throughout the text.
(11) Line No. 129: Rewrite as 'acyltransferase'
(12) Line No. 130: 'gene' is repeated twice. Delete one
(13) Figure 2: Plot error bars in the graph
(14) Figure 2: Present details of abbreviations as footnote to the figure
(15) Line No. 162 - 175: Format it 'no bold' and 'justify'
(16) Line No. 164: Rewrite as 'over expression'
(17) Line No. 218: Rewrite as 'reference genome'
(18) Discussion: Avoid presenting the results again. Present it concisely in a flow for easy understanding to the readers.
(19) Line No. 535-537: Reframe the sentence as it is not clear.
(20) Line No. 550-551: Reframe the sentence as it is not clear.
(21) Line No. 552: 'A NanoDrop'. Rewrite the sentence.
(22) Conclusion: Present the significant outcome of the study and avoid presenting the results again.
Round 2
Reviewer 4 Report
The authors have included the suggestions and made significant changes to improve the manuscript. The manuscript can be ACCEPTED for publication.
Author Response
Point 1: The authors have included the suggestions and made significant changes to improve the manuscript. The manuscript can be ACCEPTED for publication.
Answer: Thanks a million for your valuable and appreciable comments for the manuscript, “Identification of a BAHD Acyltransferase gene involved in plant growth and secondary metabolism in tea plants” (Manuscript ID: plants-1902451). We already updated this information in the new revise manuscript